# Practical Considerations for the Use of Cannabis in Cancer Pain Management—What a Medical Oncologist Should Know

**DOI:** 10.3390/jcm11175036

**Published:** 2022-08-27

**Authors:** Alecsandra Gorzo, Andrei Havași, Ștefan Spînu, Adela Oprea, Claudia Burz, Daniel Sur

**Affiliations:** 1Department of Medical Oncology, The Oncology Institute “Prof. Dr. Ion Chiricuţă”, 400015 Cluj-Napoca, Romania; 2Department of Medical Oncology, University of Medicine and Pharmacy “Iuliu Hațieganu”, 400012 Cluj-Napoca, Romania; 3Department of Allergology and Immunology, University of Medicine and Pharmacy “Iuliu Hațieganu”, 400012 Cluj-Napoca, Romania

**Keywords:** cancer pain, cannabis, Δ9-tetrahydrocannabinol, cannabidiol, opioids

## Abstract

Pain is a highly debilitating emotional and sensory experience that significantly affects quality of life (QoL). Numerous chronic conditions, including cancer, are associated with chronic pain. In the setting of malignancy, pain can be a consequence of the tumor itself or of life-saving interventions, including surgery, chemotherapy, and radiotherapy. Despite significant pharmacological advances and awareness campaigns, pain remains undertreated in one-third of patients. To date, opioids have been the mainstay of cancer pain management. The problematic side effects and unsatisfactory pain relief of opioids have revived patients’ and physicians’ interest in finding new solutions, including cannabis and cannabinoids. The medical use of cannabis has been prohibited for decades, and it remains in Schedule 1 of the Misuse of Drugs Regulations. Currently, the legal context for its usage has become more permissive. Various preclinical and observational studies have aimed to prove that cannabinoids could be effective in cancer pain management. However, their clinical utility must be further supported by high-quality clinical trials.

## 1. Introduction

Pain is an unpleasant emotional and sensory experience that is associated with potential or actual tissue damage. More than 70% of cancer patients experience pain, which impacts their emotional and physical dimensions [1]. The treatment of chronic cancer pain requires a comprehensive approach that includes both non-pharmacological and pharmacological modalities. Opioids are the foundation for managing moderate and severe cancer pain [2]. However, given the many side effects and risk of addiction, opioid-based pain management requires close monitoring [3]. According to epidemiological data, the rate of opioid overdoses has tripled since 2000, making the “opioid epidemic” one of the most challenging public health issues [4]. Although significant efforts have been made to elaborate recommendations and guidelines, many patients with cancer report inadequate pain relief using available therapeutic options. The main barriers to optimal pain management include inadequate pain assessment, fear of addiction, physicians’ reluctance to prescribe strong opioids, and inadequate access to medications [5].

In recent years, alternative pharmacological interventions for cancer pain, including cannabis-based medicines, have been widely explored. There are a series of medications available, including plant-derived cannabinoids, synthetic cannabinoids, magistral preparations of cannabis plant derivatives, nutritional supplements and experimental medications. Cannabidiol (CBD) and Δ9-tetrahydrocannabinol (THC) are the most studied compounds in the cannabis family [6]. These compounds can be administered orally as capsules or oils, via inhalation, or as a spray under the tongue or on the buccal mucosa. However, the pharmacology of cannabinoids remains limited. The various delivery systems and routes, along with the different concentrations of cannabinoids, make predicting the efficacy very challenging [7]. A series of preclinical and observational studies have tried to bring to light the possible benefit of using cannabinoids in cancer pain management. However, their clinical efficacy is still to be supported by high-quality clinical trials [8,9].

This review aims to provide an update on the use of cannabis-based medicines for cancer pain management. In addition, the current study aims to inform the medical oncology community about the use of cannabis as a possible therapeutic option for pain relief.

## 2. Overview on Cancer Pain Management, and Related Issues

Pain affects 50–90% of cancer patients and is one of the most disabling symptoms [10]. Cancer pain is classified as neuropathic or nociceptive and has a complex pathophysiology. Nociceptive pain is described as visceral or somatic pain [11]. Malignant invasion of connective tissues, skin, or bone causes somatic cancer pain and is usually characterized as a localized painful sensation. In contrast, visceral pain is often poorly localized and is caused by organ inflammation, distention, or impaction [12]. Neuropathic cancer pain can result from direct tumor invasion of nervous tissues, post-irradiation plexopathies, or chemotherapy-induced peripheral neuropathy (CIPN) [13]. Patients describe it as an electric or burning sensation, and sometimes as muscle weakness. It often manifests as persistent background pain associated with acute exacerbations and is frequently unresponsive to opioids [14]. Pain assessment usually requires the use of a visual analog scale (VAS) from 0 to 100 mm, or a numerical rating scale (NRS) from 0 to 10. In most studies, a reduction in pain intensity of more than 30%, or 20 mm on the VAS, or 2 points on the NRS is considered a clinically significant improvement [15].

Strong opioids, including morphine, are the mainstay of moderate-to-severe cancer pain management.

It is generally agreed that cancer pain management needs a comprehensive approach, including non-pharmacological and pharmacological modalities [5]. The most widely acknowledged algorithm for treating cancer pain was developed by the World Health Organization (WHO). The algorithm recommends non-opioid analgesics (acetaminophen, non-steroidal anti-inflammatory drugs) as first-line treatment. In refractory patients, the therapy should be escalated to “weak opioids” (codeine) or “strong opioids” (morphine) [16]. For patients whose pain is partially responsive to opioids, adjuvant analgesic options are available, including antidepressants, anticonvulsants, local analgesics, and corticosteroids [17].

Despite the significant progress made in cancer pain management and awareness, pain is undertreated in one-third of patients, affecting the quality of life (QoL) [18]. More than eight million people worldwide die each year of advanced cancer [19]. About six million of these patients have no or inadequate access to strong opioids because of the poor availability of these substances in the world’s most populated and impoverished countries [20].

A large study including 4707 cancer survivors showed that two-thirds of the patients reported at least one obstacle to cancer pain management. The most vulnerable groups included less-educated, non-white, and patients with more comorbidities [21]. Another prospective study including prostate, breast, colorectal, and lung cancer patients reported that minorities were twice as likely to be undertreated as white patients [22].

On the other hand, the aberrant use of opioids among patients is a serious concern in the U.S., and therefore, has led to discussions regarding epidemiological modeling and economic analysis to better allocate limited resources to attenuate the effects of the ongoing opioid epidemic. [23]. In a retrospective study, less than 50% of U.K. patients suffering from cancer received a strong opioid before their passing; however, the percentage was higher in Norway, reaching 60% [24,25]. Chronic opioid use is associated with significant side effects (constipation, nausea, vomiting, mental clouding, sleep disorders, effects on libido, and hyperalgesia), which are the main reasons for the discontinuation of analgesic treatment and QoL impairment [26]. Moreover, adverse effects often require further treatment for symptom management, leading to additional pill burden. Early provocative data from preclinical studies suggest that opioids could affect tumor progression and immune function; however, it is too premature to determine whether these data are clinically significant [27,28]. Through the endocannabinoid system, cannabis manages to regulate the immune response in different types of cells. More specifically, a series of changes appear, such as the alteration of cytokine secretion, the induction of apoptosis, and the activation of the innate and adaptive immune systems. Thus, it was observed that in patients receiving immunotherapy (immune checkpoint inhibitors), the time to tumor progression and overall survival decreased significantly [29].

Currently, an increasing number of patients with chronic cancer pain are seeking alternative treatment options. Therefore, interest in cannabis and cannabinoids has grown considerably in recent years. In addition, such products are increasingly available in many countries as the general attitude towards medical cannabis and cannabinoids has shifted [30].

## 3. Cannabis and Its Mechanisms of Action

*Cannabis sativa* L. has a long history of medicinal use. CBD and THC are the two components with the highest concentrations in *Cannabis* sp. (Figure 1) [31]. In recent years, especially since the endocannabinoid system was discovered, these compounds have received much attention [32]. The endocannabinoid system includes cannabinoid receptors (CB1 and CB2-G protein-coupled receptors), their endogenous ligands (endocannabinoids AEA—Anandamide and 2-AG—2-Arachidonoylglycerol), and the enzymes responsible for their degradation and synthesis [33]. The main enzyme involved in AEA production is N-acylphosphatidylethanolamine-specific phospholipase D (NAPE-PLD), whereas 2-AG synthesis is dependent on a specific phospholipase C followed by the sn-1-diacylglycerol lipase (DAGL) activity [34]. The AEA activity is terminated by a fatty acid amide hydrolase (FAAH) resulting in arachidonate and ethanolamine. 2-AG is hydrolyzed by a specific monoacylglycerol lipase (MAGL), as well as serine hydrolase alpha-beta-hydrolase domain 6 (ABHD6), resulting in arachidonate and glycerol [35,36]. Endocannabinoids act principally through the cannabinoid receptor (CB1 and CB2). Their implications were illustrated in several physiological and pathological conditions, including appetite, fertility, memory, immune system, cancer, and pain management [37].

CB1 receptors are primarily expressed in the peripheral and central nervous systems, while CB2 receptors are highly expressed in the immune system. Both CB1 and CB2 receptors are negatively correlated with adenylate cyclase activity [38].

A large number of cannabinoid receptors have been described in the brain stem emetic centers and in regions involved in the behavioral effects of cannabinoids, including the hippocampus, basal ganglia, amygdala, and cerebellum. CB1 receptors are located predominantly in the presynaptic membrane and are therefore modulators of synaptic release [39].

It has been assumed that cannabinoids relieve pain by activating specific CB1 and CB2 receptors. However, the matter becomes more complex when plant-derived and endogenous cannabinoids influence multiple pain targets, including G protein-coupled receptor 55 (GPCR55), GPCR18, serotonin, and opioid receptors [40,41,42,43]. Moreover, cannabinoids can modulate transient receptor potential channels (TRPA, TRPV, and TRPM), Cys-loop ligand-gated ion channels, and nuclear receptors [44]. In addition, several studies have indicated multiple interactions at the molecular level between opioids, TRPV1, and cannabinoid receptors in pain modulation and perception.

The administration of cannabinoids has been shown to suppress all neurophysiological and behavioral responses to nociceptive stimuli. These compounds were found to exert their anti-nociceptive effect by an action on the peripheral nerves, direct activity in the brain, or direct spinal activity [45]. Therefore, rapidly after crossing the brain–blood barrier, the cannabinoids can interact with the rostral ventrolateral-medulla (RVM) and periaqueductal gray (PAG), inhibiting spinal nociceptive neurotransmission [46]. Other studies have demonstrated a potential peripheral site of action for cannabinoids [47]. Hence, in a tumor-bearing mouse model, the intraplantar administration of WIN 55,212-2, a non-selective cannabinoid receptor agonist, diminished the response produced by mechanical stimulation of C-fiber nociceptors [48]. These findings could further support new drug development with improved clinical efficacy and fewer side effects [49]. The general overview of the endocannabinoid system mechanism is depicted in (Figure 2).

THC is the primary psychoactive ingredient of cannabis and can promote dependency among chronic users as it interacts with the dopaminergic system [50]. THC has an affinity for the CB1 and CB2 receptors similar to that of AEA. In addition to its psychoactive effect, THC is responsible for most of the pharmacological outcomes of cannabis, including analgesic, antioxidant, anti-inflammatory, bronchodilator, antipruritic, and anti-spastic activities [51].

CBD is the second most abundant component of cannabis and has broader medical applications than THC. However, CBD does not have THC-like or toxic drug effects but has been reported to reduce inflammation, muscle spasms, seizures, and anxiety [52]. There are studies in humans, non-human primates, and rodents that suggest the potential that CBD has in mitigating the effects of THC (mostly related to memory and behavior). There are also preclinical studies showing that CBD actually potentiates the effects of THC [53] The current state of knowledge suggests that CBD has a poor affinity for cannabinoid receptors, and its mechanism of action is different from that of the endocannabinoid system. The effects of CBD are often reported to be mediated by various orphan GPCRs and serotoninergic 5-HT1A receptors [54]. Moreover, studies have indicated that CBD can regulate pain perception by interacting with other G-coupled receptors, including δ-opioid, μ-opioid, and dopamine receptor D2 [55].

Cannabinoids are highly lipophilic substances that are stored in spleen and adipose tissues. After inhalation, peak plasma levels of CBD and THC are rapidly achieved within 3–10 min [56]. The bioavailability of inhaled THC varies considerably due to the differences between cannabis products and inhalation techniques, ranging between 10 and 35%, whereas inhaled CBD has an average bioavailability of 31%. In contrast, when administered orally, the peak plasma level of both compounds is obtained within approximately 1–2 h, and it is considerably lower than that obtained by smoking due to hepatic first-passage metabolism [57]. When inhaled by smoking, the analgesic effect of cannabinoids is experienced shortly after the first breath. However, the combustion products inhaled while smoking cannabis constitute a significant disadvantage and may negatively affect the respiratory tract [58]. On the other hand, the major limitation of oral cannabinoids is their poor pharmacokinetic profile, with highly variable absorption, slow onset of clinical effects, and unpredictable psychoactive effects [59].

The use of cannabis for recreational or medical purposes has been banned for decades. Several cannabinoid drugs have been developed to date. Nabiximols (Sativex^®^) is an almost 1:1 ratio of plant-based THC and CBD and is licensed for treating spasticity in multiple sclerosis [60]. Epidiolex, an oral CBD solution, was recently approved for the treatment of severe pediatric epilepsy, including Dravet and Lennox-Gastauld syndromes [61].

Dronabinol and nabilone are synthetic forms of THC licensed to treat chemotherapy-related nausea and vomiting in patients refractory to conventional antiemetics, and weight loss in patients with AIDS [62].

## 4. Cannabis-Based Medicines

The most documented reason for cannabinoid use is pain relief. Cancer pain is often chronic, with inflammatory, nociceptive, and neuropathic components [63]. Moreover, cancer pain is frequently challenging to control using the available therapeutic options because of its complexity. The available randomized controlled trials involving cannabinoids are limited, with equivocal results [64]. However, stimulated by the pain burden worldwide and the necessity for novel non-opioid and safer therapeutic options, interest in cannabinoids has increased in the scientific community (Table 1).

### 4.1. Preclinical Evidence

Behavioral studies have revealed that plant-based or synthetic cannabinoids may be effective in animal pain models [65]. However, the data obtained from human subjects are sometimes conflicting. Differences in methodologies and strains may explain the discrepancies observed. In humans, the perception of pain is influenced by numerous cognitive and emotional factors that can affect the results [66]. Animal studies have established cannabinoid-induced analgesia in a broad spectrum of pain models. When it comes to chronic pain, both neuropathic and inflammatory, cannabinoids have demonstrated greater potency compared to acute or physiological pain [67].

Up to 40% of the patients with cancer-related pain have neuropathic components. CB1 and CB2 receptors are both upregulated in nervous structures involved in the perception of pain caused by peripheral nerve damage, which could explain the positive effects of cannabinoid antagonists on neuropathic pain [68]. Various chemotherapeutic agents have been shown to induce CIPN in animal models, including platinum components, taxanes, vinca alkaloids, and proteasome inhibitors (bortezomib) [69]. In taxane-induced neuropathic pain, CB2-specific agonists (AM1714, R, S-AM1241, MDA7, MDA19, and AM1710) alleviate cold and mechanical allodynia via CB2 receptors. CBD has anti-nociceptive effects in paclitaxel models [70,71,72]. THC, CBD, and the CB2 agonist AM1241 alleviated vincristine-induced allodynia [73,74]. Moreover, CBD alleviated allodynia in a cisplatin model. In animal models, two endocannabinoids (AEA and 2-AG) reversed heat hyperalgesia and mechanical allodynia induced by cisplatin [75,76].

The peripheral anti-hyperalgesic effect was objectified in tissue-injury models, and it was shown that the nocifensive behavior was decreased by injecting 2-AG or AEA roughly to the injury site [77,78]. Several reports demonstrated the analgesic efficacy obtained from the pharmacological inhibition of FAAI, the main enzyme involved in AEA degradation, using carbamates, alpha-ketoheterocycle compounds, and analogs of N-arachidonoyl serotonin [79,80]. Other pre-clinical trials increased 2-AG levels by inhibiting MAGL activity and its degradation as an alternative approach [81,82]. Therefore, Khasabova et al. increased 2-AG levels mimicking its anti-hyperalgesic effect in bone cancer murine models, by administering JZL184, a selective inhibitor of MAGL [83].

### 4.2. Clinical Evidence

The first placebo-controlled trial published in 1975 reported that 15 and 20 mg THC oil provided more pain relief than placebo (*p* < 0.025) in a group of 10 patients with cancer treated with opioids (mainly methadone) [84]. A subsequent trial by the same researcher’s group included 36 patients. They concluded that the amount of pain relief produced by 10 mg of THC oil was comparable to that produced by 60 mg of codeine. However, 20 mg of THC has been reported to cause side effects such as dizziness, ataxia, somnolence, and blurred vision [85].

In addition to oils, oromucosal sprays have been widely used in clinical trials to administer cannabis-based medicines. Johnson et al. investigated, for the first time, the analgesic efficacy of mixed cannabis extracts and nabiximols (Sativex^®^) (equimolar equivalents of THC and CBD) administered orally in 177 opioid-refractory, advanced pain cancer patients. The results showed a statistically significant improvement in the mean pain score in favor of nabiximols compared with placebo (*p* < 0.024). Moreover, patients treated with nabiximols required fewer doses of breakthrough pain medication [86].

The analgesic efficacy of nabiximols was further investigated in a double-blind, randomized, placebo-controlled trial on cancer patients suffering from severe pain (numerical rating scale (NRS) scores ≥ 4 and ≤8) poorly controlled by opioids. The design implied a two-week, self-titration and tolerability phase, followed by a three-week treatment period. However, the primary efficacy endpoint was not achieved. The negative outcome of this study could be explained by several contributing factors, including the high mortality rate reported in the study population and increased dropout rate. Furthermore, self-reported NRS scores could be significantly influenced by variations in day-to-day mood, especially in frail cancer patients. Moreover, a post hoc analysis showed that Sativex was effective only in U.S. cancer patients (*p* = 0.037). The U.S. subgroup of patients had a lower opioid baseline dose and increased exposure to cannabis in the past. These findings led to the hypothesis of reduced downregulation of opioid receptors, resulting in enhanced synergy between cannabinoids and opioid receptors and, therefore, a better outcome [87].

A long-term observational study conducted in Israel evaluated the safety and efficacy of medical cannabis in 3619 cancer patients. All participants received a mixture of 16 THC and CBD strains administered with various concentrations of oils and/or inflorescences (including capsules, flowers, and cigarettes). After one month, 19.5% of the active users (2082) reported a moderate improvement, and 66.3% reported a significant improvement in their general condition. In addition, 8.3% of the patients experienced side effects, including tiredness, cough, nausea, dizziness, confusion, and disorientation. After six months, among the active users (1211), 45.1% reported moderate, and 50.8% reported significant improvements. Moreover, the percentage of patients reporting good quality of life was 68.5% compared to 18.7% at baseline (*p* < 0.001). More importantly, 9.9% of the patients reported a decrease in opioid dose, 36% discontinued opioid use, and only 1.1% increased opioid dosage. Overall, 30% of the patients reported at least one side effect: dry mouth, dizziness, sleepiness, increased appetite, and psychoactive effects [88]. In a recent systematic review and meta-analysis, non-inhaled medical cannabis and cannabinoids showed little pain relief compared to placebo in patients with chronic non-cancer and cancer pain. In addition, these products lead to minor improvements in sleep quality and physical performance [89].

Regarding neuropathic pain, Lynch et al. conducted a placebo-controlled, double-blind pilot trial that addressed chemotherapy-induced neuropathy in 18 cancer patients. Nabiximols were reported to be beneficial, with an NNT (number needed to treat) of five [90].

## 5. Existing Concerns and Legal Considerations

In addition to the limited evidence for their efficacy, cannabinoids carry the risk of adverse events, similar to any other existing therapeutic agent. Over the past 40 years, almost 5000 adverse events associated with the medical use of cannabinoids have been reported [91]. The most common side effects were psychosis, cognitive impairment, dizziness, dysphoria, dry mouth, nausea, and vomiting. Moreover, using high doses for a long time was associated with memory and psychomotor speed impairments, particularly in adolescents and young adults [92]. Additionally, cannabis use raises the likelihood of car accidents, suicidal behavior, and partner and child violence. Cannabis use is a risk factor for a number of medical disorders, as well as adverse social consequences [93]. Another major issue with cannabis-based medicine is the lack of dosing guidelines. The optimal dose (producing effective pain management with tolerable adverse effects) was inconsistent among the studies due to inter-patient variability. Moreover, side effects are difficult to assess, especially in patients with advanced cancer, who are likely to take countless concomitant medications [30,85].

The use of cannabis has been extensively debated, based on the presumption that it can lead to dependence and addiction. Statistics show that approximately 9% (1 out of 11) of all people who have ever used cannabis will experience dependency at some point in their lives, and the percentage is almost double if the use begins during adolescence [94]. It is estimated that almost 22 million people worldwide are addicted to cannabis, which is considered one of the most frequent illicit-drug-use disorders [95]. Early research implied that U.S. states with medical cannabis regulations witnessed a slower increase in opioid analgesic overdose-related mortality. However, these associations were not significant when extended time frames were analyzed [96].

The medical use of cannabis has been prohibited for many years, and it is still listed in Schedule 1 of the Misuse of Drugs Regulations. Currently, the legal context of its use is becoming more permissive. Based on the expanding evidence regarding its medical applications, the WHO proposed that cannabis should be rescheduled within international laws. Therefore, in many European countries (Figure 3), Thailand, Canada, and almost three-quarters of the U.S., medical cannabis has become legal [97,98].

## 6. Conclusions

Cancer pain is a highly debilitating syndrome that has a significant impact on QoL and is sometimes challenging to treat using the available therapeutic options. As life expectancy in cancer patients increases owing to remarkable improvements in therapies, cancer pain associated with life-saving interventions (surgery, chemotherapy, and radiotherapy) will become more prevalent. Although the results from preclinical trials are encouraging, there is a paucity of translatable evidence in clinical studies to support the use of cannabinoids for cancer pain management. Further clinical trials, more extensive and more rigorous, are needed to establish their clinical efficacy, dosing, and, not least, their potential interactions with other drugs. It is essential to carefully analyze the decision to use cannabinoids for cancer pain to avoid misuse, addiction, or dependency.

## Figures and Tables

**Figure 1 jcm-11-05036-f001:**
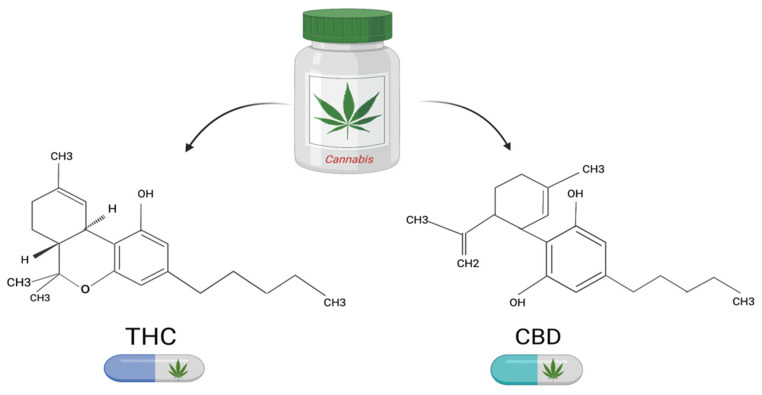
The main constituents of *Cannabis* sp. and their molecular formulas.

**Figure 2 jcm-11-05036-f002:**
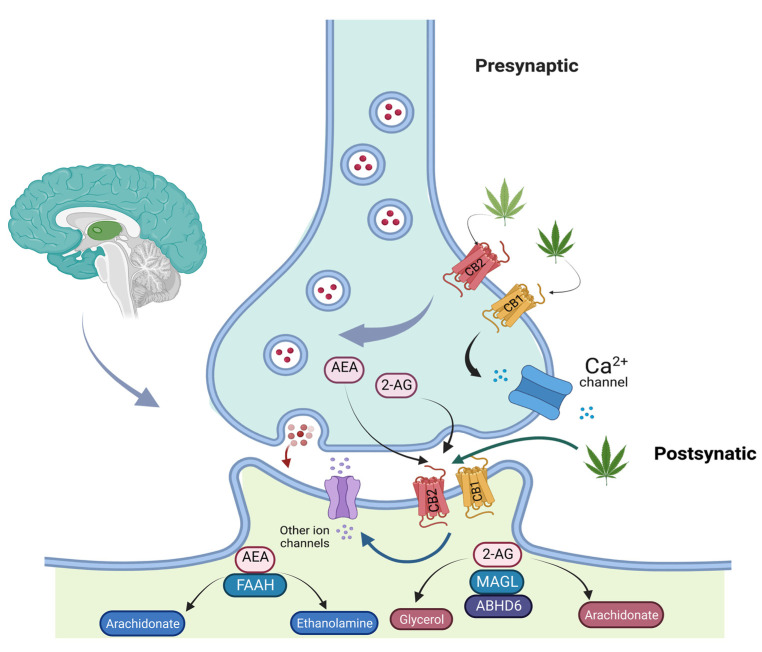
General overview of endocannabinoid system mechanism.

**Figure 3 jcm-11-05036-f003:**
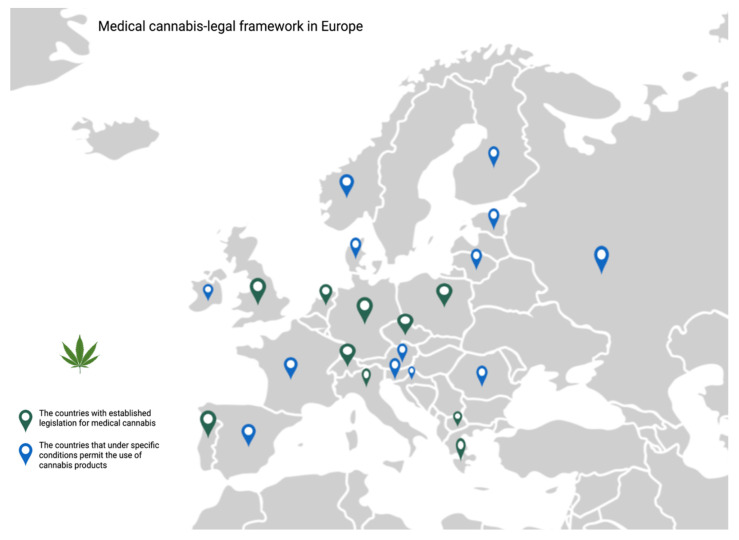
Legal framework for using medical cannabis in European countries.

**Table 1 jcm-11-05036-t001:** Ongoing clinical trials investigating Cannabinoids for cancer pain.

Study Name	Phase	Status	Condition	Treatment	Primary Endpoint
NCT04808531	Phase 3	Not yet recruiting	Cancer-related pain	NanaBis^TM^ Oxycodone Placebo Spray Placebo Tablet	Significant changes in responders with NanaBis™ spray over placebo Comparable efficacy in proportion of responders from NanaBis™ spray to the proportion of responders to Oxycodone
NCT04875286	N/A	Recruiting	Cancer-associated pain	Electronic health record review Questionnaire administration	Proportion of patients who prefer opioids + THC-marijuana and/or opioids with CBD to opioids alone for cancer pain relief
NCT03948074	Phase 2	Recruiting	Pain, nausea, anxiety, and sleep disturbance related to cancer	Cannabis	Average Patients’ Global Impression of Change (PGIC) for overall cancer-related symptoms
NCT04042545	Phase 2	Recruiting	Cancer pain QoL	Cannabis Placebo	Uncontrolled cancer pain measured using a patient’s self-administered questionnaire.

## Data Availability

Not applicable.

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
