# Peer review of "Practical Considerations for the Use of Cannabis in Cancer Pain Management—What a Medical Oncologist Should Know"

_jcm, 2022, doi:10.3390/jcm11175036_

Round 1

Reviewer 1 Report

This paper provides a nice overview of the status of cannabis-based medicines for the oncologist. The paper is well structured and follows a clear line of thought. There are a few considerations that I believe will help the reader to more clearly understand the various issues around medical cannabis and I would urge the authors to take these suggestions into consideration.

11.    Abstract, line 23.

The statement that cannabinoids “have demonstrated efficacy in cancer pain management” is too optimistic at this time. The authors conclusions at the end of the paper are much more realistic. Please edit the abstract to be in line with the conclusions.

22.    Terminology of cannabis-based medicines.

The preferred term for all medically used cannabinoids is cannabis-based medicines. It would be helpful to identify clearly at the outset the various cannabis-based medicines that are available, ie pharmacologic preparations with GMP and plant products derived from Cannabis sativa (in italics as it is the genus). There is some confusion in the text as it now stands,  eg, line 43 Cannabis Sativa-based medicines (incorrect, it is either nabiximols, or the plant Cannabis Sativa. Line 179 change “cannabis for cancer pain” to cannabis-based medicines.

33.   Organization of preclinical evidence and clinical evidence.

It would be helpful to provide a section labelled “preclinical evidence” followed by a section “clinical evidence”. Eg Line245-251 at the end of the section on cannabis for cancer pain describes preclinical animal models which could lead to confusion.

44. Ref 23 does not address “tightening of opioid prescription regulations”, Line 92, but is rather a discussion regarding epidemiological modelling and economic analysis to better allocate limited resources to attenuate the effects of the ongoing opioid epidemic. Please correct.

55.  Some English corrections

a.       Line 26 “invalidating symptoms” should probably read “most disabling”

b.       Line 122 “brain streams emetic”  should read brain stem.

c.       Line 140 syntax is incorrect…. Should read “THC has an affinity for the CB1 and CB2 receptors similar to that of AEA”

d.       Line 189 “species” is incorrect….should be “strains”

e.       Line 211…endpoint was not determined….should read was not achieved

66.  Line 147 “CBD can reduce the psychoactive properties of THC……”

this is not fully accepted, and this statement should be tempered in line with current evidence…see Niesink R, van Laar M 2013,  and Boggs et al 2017 Neuropsychopharmacology.

77.  Entourage effect

This term refers to the combination of the many various molecules in C. sativa including terpenes, flavonoids…and not just CBD….and refers to a suggested additive effect of various molecules to achieve a positive effect. Please revise.

88.  Please reference the recent paper on cannabis and check point inhibitors which is important for oncologists. Bar-Sela G et al Cannabis consumption used by cancer patients during immunotherapy correlates with poor clinical outcome, Cancers (Basel) 2020.

99.  Cannabinoids are safe in controlled setting, line 260. Ref 76. This statement is incorrect and reference refers to legalization of non-medical cannabis. There is ample evidence for considerable risks related to use of cannabinoids in general. Please reference Campeny E Eur Neuropsychopharmacol, April 2020, systematic review of systematic reviews of cannabis use   related health harms…..44 systematic reviews of 1053 studies with evidence for clear association of many harms

Author Response

Title: Practical Considerations For The Use Of Cannabis In Cancer Pain Management – What A Medical Oncologist Should Know

Authors: Alecsandra Gorzo, Andrei HavaÈ™i, Ștefan Spînu, Adela Oprea, Claudia Burz and Daniel Sur

We want to thank the reviewer for the time allocated to analyze our manuscript. We are pleased to know our review’s content was appreciated. Furthermore, we are convinced that we will improve the current article by answering the editor's requests. We have amended our manuscript according to the reviewer’s suggestions.

Reviewer’s comments

This paper provides a nice overview of the status of cannabis-based medicines for the oncologist. The paper is well structured and follows a clear line of thought. There are a few considerations that I believe will help the reader to more clearly understand the various issues around medical cannabis and I would urge the authors to take these suggestions into consideration.

Thanks for taking the time to review this article. We appreciate the encouragement and suggestions for improvement that you have given us, and in the following we will respond punctually for each individual suggestion.

  1. 11.   Abstract, line 23.

The statement that cannabinoids “have demonstrated efficacy in cancer pain management” is too optimistic at this time. The authors conclusions at the end of the paper are much more realistic. Please edit the abstract to be in line with the conclusions.

 R: We have reformulated the phrase to be in line with the introduction and the conclusion.

  1.  Terminology of cannabis-based medicines.

The preferred term for all medically used cannabinoids is cannabis-based medicines. It would be helpful to identify clearly at the outset the various cannabis-based medicines that are available, ie pharmacologic preparations with GMP and plant products derived from Cannabis sativa (in italics as it is the genus). There is some confusion in the text as it now stands,  eg, line 43 Cannabis Sativa-based medicines (incorrect, it is either nabiximols, or the plant Cannabis Sativa. Line 179 change “cannabis for cancer pain” to cannabis-based medicines.

 R: We have added in the text the various cannabis-based medicines that are available at this moment, and we have corrected the title ”cannabis-based medicines” in accordance with your suggestion.

  1. 33. Organization of preclinical evidence and clinical evidence.

It would be helpful to provide a section labelled “preclinical evidence” followed by a section “clinical evidence”. Eg Line245-251 at the end of the section on cannabis for cancer pain describes preclinical animal models which could lead to confusion.  

R: We amended our manuscript according to the reviewer's suggestion, and we included two subsections, "preclinical evidence" and "clinical evidence"

  1. Ref 23 does not address “tightening of opioid prescription regulations”, Line 92, but is rather a discussion regarding epidemiological modelling and economic analysis to better allocate limited resources to attenuate the effects of the ongoing opioid epidemic. Please correct.

 R: We have corrected the text according to the data found in reference 23.

  1. Some English corrections
  2. Line 26 “invalidatingsymptoms” should probably read “most disabling”
  3. Line 122 “brain streamsemetic”  should read brain stem.
  4. Line 140 syntax is incorrect…. Should read “THC has an affinity for the CB1 and CB2 receptors similar to that of AEA”
  5. Line 189 “species” is incorrect….should be “strains”
  6. Line 211…endpoint was not determined….should read was not achieved

R: Thank you for the corrections you offered us regarding the English language used in the text. 

  1. Line 147 “CBD can reduce the psychoactive properties of THC……”

this is not fully accepted, and this statement should be tempered in line with current evidence…see Niesink R, van Laar M 2013,  and Boggs et al 2017 Neuropsychopharmacology.

R: We deleted the initial phrase and reformulated it according to Boggs et al 2017 Neuropsychopharmacology as reference.

  1. Entourage effect

This term refers to the combination of the many various molecules in C. sativa including terpenes, flavonoids…and not just CBD….and refers to a suggested additive effect of various molecules to achieve a positive effect. Please revise.

R: We have given up using this term.

  1. Please reference the recent paper on cannabis and check point inhibitors which is important for oncologists. Bar-Sela G et al Cannabis consumption used by cancer patients during immunotherapy correlates with poor clinical outcome, Cancers (Basel) 2020.

R: We have added to the text data about the use of cannabis during immunotherapy according to the existing evidence.

  1. Cannabinoids are safe in controlled setting, line 260. Ref 76. This statement is incorrect and reference refers to legalization of non-medical cannabis. There is ample evidence for considerable risks related to use of cannabinoids in general. Please reference Campeny E Eur Neuropsychopharmacol, April 2020, systematic review of systematic reviews of cannabis use   related health harms…..44 systematic reviews of 1053 studies with evidence for clear association of many harms

R: We revised the sentence introducing updated data according to Campeny E Eur Neuropsychopharmacol, April 2020

To conclude, we want to address our gratitude for reviewing our manuscript.

Reviewer 2 Report

This is a clearly written paper that describes clinical studies related to benefits of cannabinoids for cancer pain.  Clinical studies on efficacy and side effects are clearly described.  Also, it is appropriately pointed out that there is a need for well-controlled clinical studies.  There are a few suggestions for improvement.

1.  There is no discussion of the possibility of increasing levels of  endocannabinoids, by blocking degradation, as a possible treatment for cancer pain.  Several pre-clinical studies have been published and should be referenced (e.g. Khasabova et al., 2011).

2.  In this regard, figure 2 shows an overview of the endocannabinoid system but this is simplistic.  The endocannabinoids are not shown, only their receptors.  This, or another figure, should show the metabolic pathways for the endocannabinoids Anandamide and 2-AG.

3.  There is no mention of how the cannabinoids are reducing cancer pain.  Pre-clinical mouse models have shown that cannabinoids decrease excitability and tumor-induced sensitization of dorsal horn neurons.  Importantly, cannabinoids were shown to decrease sensitization of C-fiber nociceptors in a mouse cancer pain model.  This is particularly interesting because it suggests that peripherally-restricted cannabinoids might be effecting in resucing cancer pain without the side effects associated with activation of cannabinoid receptors in the CNS.  This should be discussed and references provided.

Author Response

Title: Practical Considerations For The Use Of Cannabis In Cancer Pain Management – What A Medical Oncologist Should Know

Authors: Alecsandra Gorzo, Andrei HavaÈ™i, Ștefan Spînu, Adela Oprea, Claudia Burz and Daniel Sur

We want to thank the reviewer for the time allocated to analyze our manuscript. We are pleased to know our review’s content was appreciated. Furthermore, we are convinced that we will improve the current article by answering the editor's requests. We have amended our manuscript according to the reviewer’s suggestions.

Reviewer’s comments

This is a clearly written paper that describes clinical studies related to benefits of cannabinoids for cancer pain.  Clinical studies on efficacy and side effects are clearly described.  Also, it is appropriately pointed out that there is a need for well-controlled clinical studies.  There are a few suggestions for improvement.

  1. There is no discussion of the possibility of increasing levels of  endocannabinoids, by blocking degradation, as a possible treatment for cancer pain.  Several pre-clinical studies have been published and should be referenced (e.g. Khasabova et al., 2011).

We amended our manuscript according to the reviewer’s suggestion. Also, for a better understanding, we included additional information about the endocannabinoid system. Now the text can found as:

“The main enzyme involved in AEA production is N-acylphosphatidylethanolamine-specific phospholipase D (NAPE-PLD), whereas 2-AG synthesis is dependent on a specific phospholipase C followed by the sn-1-diacylglycerol lipase (DAGL) activity[34]. The AEA activity is terminated by a fatty acid amide hydrolase (FAAH) resulting arachidonate and ethanolamine. 2-AG is hydrolyzed by a specific monoacylglycerol lipase (MAGL) as well as serine hydrolase alpha-beta-hydrolase domain 6 (ABHD6) resulting arachidonate and glycerol[35,36]. Endocannabinoids act principally through the cannabinoid receptor (CB1 and CB2). Their implications were illustrated in several physiological and pathological conditions, including appetite, fertility, memory, immune system, cancer, and pain management[37].”

And

“The peripheral anti-hyperalgesic effect was objectified in tissue-injury models, and it was shown that the nocifensive behavior was decreased by injecting 2-AG or AEA roughly to the injury site[76,77]. Several reports demonstrated the analgesic efficacy obtained from the pharmacological inhibition of FAAI, the main enzyme involved in AEA degradation, using carbamates, alpha-ketoheterocycle compounds, and analogs of N-arachidonoyl serotonin[78,79]. Other pre-clinical trials increased 2-AG levels by inhibiting MAGL activity and its degradation as an alternative approach[80,81]. Therefore, Khasabova et al. increased 2-AG levels mimicking its anti-hyperalgesic effect in bone cancer murine models, by administering JZL184, a selective inhibitor of MAGL[82].”

  1. In this regard, figure 2 shows an overview of the endocannabinoid system but this is simplistic.  The endocannabinoids are not shown, only their receptors.  This, or another figure, should show the metabolic pathways for the endocannabinoids Anandamide and 2-AG.

We modified figure 2 according to the reviewer’s suggestion. Now it appears like:

  1. There is no mention of how the cannabinoids are reducing cancer pain.  Pre-clinical mouse models have shown that cannabinoids decrease excitability and tumor-induced sensitization of dorsal horn neurons.  Importantly, cannabinoids were shown to decrease sensitization of C-fiber nociceptors in a mouse cancer pain model.  This is particularly interesting because it suggests that peripherally-restricted cannabinoids might be effecting in resucing cancer pain without the side effects associated with activation of cannabinoid receptors in the CNS.  This should be discussed and references provided.

We amended our manuscript according to the reviewer’s suggestion. Now the text can be read as: “ The administration of cannabinoids was shown to suppress all neurophysiological and behavioral responses to nociceptive stimuli. These compounds were found to exert their anti-nociceptive effect by an action on the peripheral nerves, direct activity in the brain, or direct spinal activity[45]. Therefore, rapidly after crossing the brain-blood barrier, the cannabinoids can interact with the rostral ventrolateral-medulla (RVM) and periaqueductal gray (PAG), inhibiting the spinal nociceptive neurotransmission[46]. Other studies demonstrated a potential peripheral site of action for cannabinoids[47]. Hence, in a tumor-bearing mice model, the intraplantar administration of WIN 55,212-2, a non-selective cannabinoid receptor agonist, diminished the response produced by mechanical stimulation of C-fiber nociceptors[48].”

To conclude, we want to address our gratitude for reviewing our manuscript.

Round 2

Reviewer 2 Report

The authors have responded well to the previous concerns and have made appropriate modifications and edits to the manuscript.  The manuscript is improved and more comprehensive.  There are no additional concerns or suggestions.